# Towards Accurate Deep Learning Model Selection: A Calibrated Metric Approach

## Abstract

The adoption of deep learning across various fields has been extensive, yet the methods for reliably evaluating the performance of deep learning pipelines remain underdeveloped. Typically, with the increased use of large datasets and complex models, the training process is run only once and the new modeling result is compared to previous benchmarks. This practice can lead to imprecise comparisons due to the variance in deep learning pipelines, which stems from the inherent randomness in the training process. Traditional solutions often require running the training process multiple times and are often infeasible in Deep Learning due to computational constraints. In this paper, we introduce a calibrated metric approach, designed to address this issue by reducing the variance present in its conventional counterpart. Consequently, this new metric improves the accuracy in detecting effective modeling improvements in the model selection stage. The efficacy of the new approach has been justified both theoretically and empirically.

## 1 Introduction

The progress in machine learning is largely influenced by experimental outcomes, particularly in the era of deep learning. Researchers often evaluate the performance of new methods by comparing them with previous benchmark results to demonstrate the superiority of new methods. However, it is well known that the performance of deep learning models can vary greatly, even when using the same pipeline (Picard, 2021; Pham et al., 2020; Reimers & Gurevych, 2017), where, in this work, we define the pipeline broadly, which includes but is not limited to the selection of feature sets, model architectures, optimization algorithms, initialization schemes, and hyperparameters. Two identical pipelines may produce substantially different validation metrics due to factors such as random initialization, data shuffle, and optimization noise. This variability makes it difficult to accurately compare the modeling improvements over previous baselines. Even significant engineering efforts may only lead to small measured gains within the noise margin.

In fact, it has been shown that by selecting a lucky random initialization seed, one can achieve a model performance that is significantly better than average (Picard, 2021). This difference can be substantial enough to be used as a strong argument for publications in selective venues (Picard, 2021).

This issue is exacerbated in industry, where the production model performance is hard to improve, while there are hundreds of machine learning engineers working on the same model at the same time. The performance gain of a modeling proposal is usually small and within the metric variance, making it difficult to judge the effectiveness of the modeling proposal.

To address this issue, a common approach is to run the training pipeline multiple times and report average, standard deviation, minimum, and maximum performance scores (Picard, 2021). Reimers & Gurevych (2017) proposed to report the score distributions based on multiple executions instead of single performance scores. However, with the rise of large training data and large models, these approaches are not always practical due to limited computational resources (Bouthillier & Varoquaux, 2020).

In this work, we present a new perspective that focuses on metric design. The key insight is that we can reduce the variance of the evaluation metrics themselves to enable more accurate comparisons between models. We propose a metric framework called Calibrated Metric that exhibits lower variance than its counterpart. Our method works by correcting for inherent biases before finalizing

the metric calculations on a holdout set. Our method does not require additional validation data and is easy to compute.

We provide theoretical justifications for our metric in a linear regression setting. We demonstrate the effectiveness of our method empirically in binary classification tasks and regression tasks. Experiments on real-world data show that our Calibrated Metric reliably detects superior models compared to its counterpart. We further validate the benefits under different deep learning training configurations.

In summary, this paper makes the following contributions:

- We formulate the deep learning pipeline evaluation problem and propose to tackle it by designing new metrics.

- We propose a new metric framework, Calibrated Metric, which can mitigate the above deep learning pipeline evaluation issue.

- We conduct extensive experiments to demonstrate the effectiveness of the proposed metric, using both synthetic datasets and real-world datasets.

- We provide theoretical guarantees in a linear regression setting that the proposed metric has a smaller variance than its vanilla counterpart.

## 2 PRELIMINARIES AND PROBLEM SETTING

In this section, we examine the supervised learning setting, where we assume that the training data, validation data, and test data are randomly drawn from an unknown distribution in an i.i.d. manner, denoted as $\mathcal{D}$. Our work can naturally be generalized to the concept drift[1] setting, which we will discuss in detail in Section 5.5.

Our goal is to develop a good pipeline that maps from a training distribution to a possibly random model $h \in \mathcal{H}$, which generalizes well during the test time. As we mentioned in the Introduction 1, the pipeline incorporates the whole procedure for training a model, including the selection of model architectures, optimization algorithms, initialization schemes, and hyperparameters. Model performance is evaluated by a metric, $e$, and thus the expected performance of a model $h$ is

$$R_e(h) = \mathbb{E}_{\mathcal{D}}[e(h(X), Y)|h]. \tag{1}$$

In practice, $R_e(h)$ is estimated by the finite-sample average on the test data set $\hat{\mathcal{D}}_{\text{test}}$. That is,

$$\hat{R}_e(h, \hat{\mathcal{D}}_{\text{test}}) = \frac{1}{|\hat{\mathcal{D}}_{\text{test}}|} \sum_{(x,y) \in \hat{\mathcal{D}}_{\text{test}}} e(h(x), y). \tag{2}$$

It should be noted that the expected risk, $R_e(h)$, is a random variable, since $h$ is random and depends on a specific model that is produced by the underlying deep learning pipeline. The output model is random due to the randomness of the data from the sample collection and intrinsic randomness during the training process in the deep learning pipeline, such as the order of the data and the randomness of the descent of the stochastic gradient. Therefore, a proper evaluation and comparison of different deep learning pipelines should take into account the distribution of $R_e(h)$ (Bouthillier et al., 2021; Reimers & Gurevych, 2017). It is also important to note that the term "deep learning pipeline" in this context is general, as we consider different model configurations (e.g. different model hyperparameters) as different "deep learning pipelines", even though they may belong to the same model class.

To compare the performance of different deep learning pipelines, we should calculate the expected risk $R_e(h)$ for each pipeline. As mentioned above, such expected risk is a random variable w.r.t. $h$, then we should compare the distribution of $R_e(h)$. Specifically, we use the probability that the expected risk $R_e(h)$ for one pipeline is larger or smaller than that of the other to quantify the pairwise performance comparison between different pipelines.

---

[1]Concept Drift refers to unforeseeable changes in the underlying distribution of streaming data overtime (Lu et al., 2018).

**Definition 2.1.** *(Better pipeline)* For any two pipelines $A$ and $B$, we say that pipeline $A$ is better than pipeline $B$ with respect to metric $e$ if and only if the probability that pipeline $A$ produces a better model (i.e., smaller risk), measured by the metric $e$, is greater than $0.5$. This is represented by the inequality:

$$\mathbb{P}\left(R_e(h_A) < R_e(h_B)\right) > 0.5 \tag{3}$$

where $h_A$ and $h_B$ are random variables representing the output models produced by pipeline $A$ and $B$ respectively.

Our objective is to compare the performance of two pipelines, $A$ and $B$, with respect to the metric $e$ by running the training pipeline only once. Ideally, the Monte Carlo method could be used to estimate $\mathbb{P}\left(R_e(h_A) < R_e(h_B)\right)$, but this approach demands substantial computational resources, making it impractical in Deep Learning. In this work, we aim to come up with an alternative metric $e_1$ with the following properties:

1. *Roughly same mean*
$$\mathbb{E}\left[R_{e_1}(h)\right] \approx \mathbb{E}\left[R_e(h)\right];$$

2. *Strictly small variance*
$$var\left(R_{e_1}(h)\right) < var\left(R_e(h)\right),$$

where the randomness is from the pipeline that produces $h$. As a result, the new metric can compare the performance of pipelines $A$ and $B$ more accurately using the same computational resources.

**Definition 2.2.** *(Better alternative metric)* Assuming that pipeline $A$ is better than $B$ with respect to the metric $e$ (i.e. pipeline $A$ is more likely to produce a better model than pipeline $B$ in the ground truth if measured by metric $e$), we say that a metric $e_1$ is better than $e_2$ with respect to metric $e$ if and only if the probability that pipeline $A$ produces a better model than pipeline $B$ measured by metric $e_1$ is greater than the probability measured by metric $e_2$. This is represented by the inequality:

$$\mathbb{P}\left(R_{e_1}(h_A) < R_{e_1}(h_B)\right) > \mathbb{P}\left(R_{e_2}(h_A) < R_{e_2}(h_B)\right) \tag{4}$$

In other words, using metric $e_1$ is more likely to accurately detect that pipeline $A$ is better than pipeline $B$, which aligns with the ground truth. Here, we allow for a general form of the risk function, which may not admit the expectation form; i.e., $R_{e_1}(h)$ may not necessarily have the form $\mathbb{E}_{\mathcal{D}}[e_1(h(X), Y)]$.

**Definition 2.3.** *(Metric accuracy)* We assume without loss of generality that pipeline $A$ is better than $B$ with respect to the metric $e$. We define the accuracy of a metric $\bar{e}$ with respect to metric $e$ and pipeline $A$ and $B$ as:

$$\text{Acc}(\bar{e}) \triangleq \mathbb{P}\left(R_{\bar{e}}(h_A) < R_{\bar{e}}(h_B)\right) \tag{5}$$

Our goal is to find a metric $\bar{e}$ that has a higher accuracy than the original metric $e$ for a wide range of pipelines $A$ and $B$. In the next section, we will present a new metric framework, namely Calibrated Metric. The intuition is that the bias in the function $h$ is always volatile and carries on a great deal of randomness. Calibrating the bias will usually not change the comparison between two pipelines but can reduce the randomness. In Section 4, we will present a theoretical analysis that justifies this intuition by showing that our new metric framework has a smaller variance in the linear regression setting. Through extensive experiments in Section 5, we will show that Calibrated Metric achieves higher accuracy than its counterpart for a wide range of tasks and deep learning pipelines.

## 3 CALIBRATED METRIC FRAMEWORK

**Overview of the Framework** Our framework is outlined below in Algorithm 1. It has three main steps:

1. Partition $\hat{\mathcal{D}}_{\text{val}}$ into $\hat{\mathcal{D}}_{\text{val}-\text{bias}}$ and $\hat{\mathcal{D}}_{\text{val}-\text{remain}}$.

2. Use model predictions $p_i^{\text{val}-\text{bias}}$ and labels $y_i^{\text{val}-\text{bias}}$ to compute bias term $c^*$.

3. Apply the bias term $c^*$ to $p_i^{\text{val}-\text{remain}}$ to obtain bias-adjusted predictions $q_i^{\text{val}-\text{remain}}$, and compute corresponding metric using $q_i^{\text{val}-\text{remain}}$ and labels $y_i^{\text{val}-\text{remain}}$.

---

**Algorithm 1** Calculate Calibrated Metric

---

1: **Input:** model $h$, labeled validation data $\hat{\mathcal{D}}_{\text{val}}$, vanilla metric $e$, bias correction formula $f_e$.
2: **Output:** Calibrated Metric: $\hat{R}_{e_1}(h, \hat{\mathcal{D}}_{\text{val}})$.
3: Partition $\hat{\mathcal{D}}_{\text{val}}$ into $\hat{\mathcal{D}}_{\text{val-bias}}$ and $\hat{\mathcal{D}}_{\text{val-remain}}$.
4: Compute model predictions on $\hat{\mathcal{D}}_{\text{val-bias}}$ and $\hat{\mathcal{D}}_{\text{val-remain}}$, denoted as $p_i^{\text{val-bias}}$ and $p_i^{\text{val-remain}}$; Labels are denoted as $y_i^{\text{val-bias}}$ and $y_i^{\text{val-remain}}$.
5: Compute bias term $c^*$ using $p_i^{\text{val-bias}}$ and $y_i^{\text{val-bias}}$.
6: Calculate bias-adjusted predictions $q_i^{\text{val-remain}}$ using formula $q_i = f_e(p_i, c^*)$ .
7: Calculate the Calibrated Metric $\hat{R}_{e_1}(h, \hat{\mathcal{D}}_{\text{val}})$ by applying the Vanilla Metric $e$ to $q_i^{\text{val-remain}}$ and $y_i^{\text{val-remain}}$.

---

**Calibrated Log Loss Metric (Binary Classification)** The Calibrated Log Loss Metric is useful when the Log Loss Metric is commonly used as a core metric to evaluate model performance. A typical application that utilizes the Log Loss Metric as its primary evaluation criterion is the Click-Through Rate (CTR) prediction task (He et al., 2014; Wang et al., 2017; McMahan et al., 2013).

In the field of Deep Click-Through Rate Prediction Models, it is common for models to overfit when trained for more than one epoch (Zhou et al., 2018; Zhang et al., 2022). As a result, models are often trained only for a single epoch in practice (Zhang et al., 2022), making it uncertain whether the model has been fully optimized. This leads to volatility of the bias term in the final layer of neural networks, creating additional randomness.

Let $\text{logit}(p) := \log(\frac{p}{1-p})$ and $g^c(p) := (1 + e^{-\text{logit}(p)+c})^{-1}$.

To compute $c^*$, the following convex optimization program is solved:

$$
c^* = \min_c \left\{ - \sum_{\substack{i \in \\ \hat{\mathcal{D}}_{\text{val-bias}}}} (y_i \log(g^c(p_i)) + (1 - y_i) \log(1 - g^c(p_i))) \right\}. \tag{6}
$$

Let $q_i = f_e(p_i, c^*) := g^{c^*}(p_i)$. It can be easily shown that the bias-adjusted predictions $q_i$ are well calibrated in $\hat{\mathcal{D}}_{\text{val-bias}}$, which means that $\sum_{i \in \hat{\mathcal{D}}_{\text{val-bias}}} q_i = \sum_{i \in \hat{\mathcal{D}}_{\text{val-bias}}} y_i$.

The Calibrated Log Loss metric is

$$
\hat{R}_{e_1}(h, \hat{\mathcal{D}}_{\text{val}}) = \frac{1}{|\hat{\mathcal{D}}_{\text{val-remain}}|} \sum_{\substack{i \in \\ \hat{\mathcal{D}}_{\text{val-remain}}}} e(q_i, y_i),
$$

and

$$
e(p, y) = y \log(p) + (1 - y) \log(1 - p).
$$

**Calibrated Quadratic Loss Metric (Regression)** The Calibrated Quadratic Loss Metric is useful when the Mean Squared Error (MSE) Metric is commonly used as a core metric to evaluate model performance. A typical application that uses MSE as its primary evaluation criterion is the stock return prediction task (Jiang, 2021; Hu et al., 2021; Zou et al., 2022).

$c^*$ is easy to compute in the Calibrated Quadratic Loss Metric case:

$$
c^* = \frac{1}{|\hat{\mathcal{D}}_{\text{val-bias}}|} \sum_{\substack{i \in \\ \hat{\mathcal{D}}_{\text{val-bias}}}} (y_i - p_i).
$$

And $f_e(p_i, c^*) = p_i + c^*$.

**Additional Computational Cost of the Framework** The Calibrated Metric introduces minimal overhead, and the associated computational cost is often negligible. This is because the bias calculation in step 2 is highly efficient, and computing the bias-adjusted predictions in step 3 requires only a few floating-point operations.

## 4 THEORY ON LINEAR REGRESSION

In this section, we provide theoretical justification that our new metric has a smaller variance than its vanilla counterpart under Linear Regression setting, where the randomness only comes from the data randomness. We choose to provide a theoretical guarantee under Linear Regression due to its simplicity. We empirically verify our method's performance under Linear Regression, Logistic Regression, and Neural Networks in the next section. Note that in Linear Regression, the Quadratic Loss Metric is used.

**Theorem 4.1.** *Suppose that the features $X \in \mathbb{R}^d$ and the label $Y$ are distributed jointly Gaussian. We consider linear regression $h(x) = \beta^\top x + \alpha$. Let $\hat{\beta}_n$ be the coefficient learned from the training data with sample size $n$. Then, we have*

$$\left(1 + \frac{1}{n}\right) \mathbb{E}[e_1(h(X), Y)|\hat{\beta}_n] = \mathbb{E}[e(h(X), Y)|\hat{\beta}_n],$$

*where the expectation is taken over the randomness over both the training and test samples,*

$$e(h(x), y) = (y - h(x))^2, \text{ and}$$

$$e_1(h(x), y) = (y - h(x) - (\mathbb{E}_{\mathcal{D}}[Y] - \mathbb{E}_{\mathcal{D}}[h(X)|h]))^2.$$

Let $\hat{\alpha}_n$ be the learned intercept. Note that the original risk and the calibrated risk are

$$R_e(h) = \mathbb{E}[e(h(X), Y)|\hat{\beta}_n, \hat{\alpha}_n], \text{ and}$$

$$R_{e_1}(h) = \mathbb{E}[e_1(h(X), Y)|\hat{\beta}_n, \hat{\alpha}_n] = \mathbb{E}[e_1(h(X), Y)|\hat{\beta}_n].$$

Therefore, Theorem 4.1 implies that

$$(1 + \frac{1}{n})\mathbb{E}[R_{e_1}(h)] = \mathbb{E}[R_e(h)].$$

Furthermore, to make $e$ and $e_1$ comparable, we should scale $e_1$ to $(1 + \frac{1}{n})e_1$. We show that after scaling, $(1 + \frac{1}{n})R_{e_1}(h)$ has a smaller variance than $R_e(h)$ in the next corollary. In practice, since $(1 + \frac{1}{n})$ is a constant as long as the training sample size is fixed, we can directly compare two pipelines using $R_{e_1}(h)$.

**Corollary 4.2.** *Suppose that $h_1(x)$ and $h_2(x)$ are two different learned linear functions in different feature sets. Then, we have*

$$\mathbb{E}[R_e(h_1)] = \mathbb{E}[R_e(h_2)] \Leftrightarrow \mathbb{E}[R_{e_1}(h_1)] = \mathbb{E}[R_{e_1}(h_2)] \tag{7}$$

*and*

$$var\left(\left(1 + \frac{1}{n}\right) R_{e_1}(h)\right) < var(R_e(h))$$

*for any $h$ learned from linear regression.*

Corollary 4.2 indicates that Calibrated Quadratic Loss Metric has a smaller variance than vanilla Quadratic Loss Metric without changing the mean after appropriate scaling. Note that smaller variance and higher accuracy (inequality 4) are highly correlated under mild conditions, but a smaller variance alone does not guarantee higher accuracy. In the next section, we will empirically demonstrate that the new metric has a smaller variance and achieves higher accuracy. All proofs can be found in the Appendix A.

## 5 EXPERIMENT RESULTS

### 5.1 ESTIMATION OF ACCURACY

Recall that accuracy of a metric $\bar{e}$ is defined as:

$$\text{Acc}(\bar{e}) \triangleq \mathbb{P}(R_{\bar{e}}(h_A) < R_{\bar{e}}(h_B)).$$

To obtain an estimate of $\text{Acc}(\bar{e})$, we run pipelines $A$ and $B$ for $m$ times, obtaining models $h_{A_i}$ and $h_{B_i}$ for $i \in [m]$. $\text{Acc}(\bar{e})$ can be estimated as:

$$\widehat{\text{Acc}}(\bar{e}) = \frac{1}{m^2} \sum_{(i,j)} \mathbb{1}(\hat{R}_{\bar{e}}(h_{A_i}, \hat{\mathcal{D}}_{\text{val}}) < \hat{R}_{\bar{e}}(h_{B_j}, \hat{\mathcal{D}}_{\text{val}})) \tag{8}$$

$\widehat{\text{Acc}}(\bar{e})$ is an unbiased estimator of $\text{Acc}(\bar{e})$, and in the experiments below, we report $\widehat{\text{Acc}}(\bar{e})$ as our accuracy metric. In all the tables in this section, without loss of generality, we write the tables as pipeline A is better than pipeline B in the sense of $\mathbb{P}(R_e(h_A) < R_e(h_B)) > 0.5$.

## 5.2 Synthetic Data

In Appendix B.2, we consider a linear regression model to provide empirical evidence to support our theory in Calibrated Quadratic Loss Metric setting. We also consider a logistic regression model to demonstrate the effectiveness of the Calibrated Log Loss Metric. All details and results can be found in the Appendix B.2.

## 5.3 Avazu CTR Prediction dataset

**Dataset** The Avazu CTR Prediction dataset [2] is a common benchmark dataset for CTR predictions. Due to computational constraints in our experiments, we use the first 10 million samples, randomly shuffle the data set, and split the whole data set into 80% $\hat{\mathcal{D}}_{\text{train}}$, 2% $\hat{\mathcal{D}}_{\text{val-bias}}$, and 18% $\hat{\mathcal{D}}_{\text{val-remain}}$.

**Metrics** We compare the accuracy of Calibrated Log Loss Metric and Log Loss Metric. To make a fair comparison, we compute Log Loss metric on $\hat{\mathcal{D}}_{\text{val}} = \hat{\mathcal{D}}_{\text{val-bias}} + \hat{\mathcal{D}}_{\text{val-remain}}$.

**Base Model** We use the open source xDeepFM model (Lian et al., 2018) implemented in Shen (2017) as our base model. We primarily conduct experiments using xDeepFM models, including hyperparameter-related experiments and feature-related experiments. To demonstrate that our new metric can also handle comparisons between different model architectures, we also conduct experiments using DCN (Wang et al., 2017), DeepFM (Guo et al., 2017), FNN (Zhang et al., 2016), and DCNMix (Wang et al., 2021).

**Experiment Details** We consider neural networks with different architectures, different training methods, different hyper-parameters, and different levels of regularization as different pipelines. Such comparisons represent common practices for research and development in both industry and academia. For each pipeline, we train the model 60 times with different initialization seeds and data orders to calculate $\widehat{\text{Acc}}(\bar{e})$. Note that we use "Log Loss Metric" as our ground truth metric to determine the performance rank of different pipelines. Due to computational constraints, we cannot afford to run the experiments for multiple rounds. Instead, we run the experiments for one round and report accuracy. Note that in the neural network experiments, we do not re-sample the training data each time, as there is intrinsic randomness in the neural network training process (Lakshminarayanan et al., 2016; Lee et al., 2015). This is the main difference from the Linear Regression and Logistic Regression experiments.

**Pipelines with Different Number of Features** In this set of experiments, for pipeline $A$, we use all the features available. For pipeline $B$, we remove some informative features. We tested the removal of 6 dense features and 1 sparse features respectively.

Table 1: Accuracy of Log Loss Metric (LL) and Calibrated Log Loss Metric (CLL) (features)

| Pipeline A | Pipeline B | LL Acc | CLL Acc |
|---|---|---|---|
| Baseline | remove dense | 81.8% | 88.8% |
| Baseline | remove sparse | 78.6% | 85.9% |

From the result in Table 1, we can clearly see that Calibrated Log Loss Metric has a higher accuracy, indicating its effectiveness when comparing the performance of pipelines with different features.

---

[2] https://www.kaggle.com/c/avazu-ctr-prediction

Table 2: Mean and Standard Deviation of Log Loss Metric (LL) and Calibrated Log Loss Metric (CLL) (features)

| Pipeline | LL Mean | CLL Mean | LL Std | CLL Std |
|---|---|---|---|---|
| remove dense | 0.37408 | 0.37403 | $4.7 \times 10^{-4}$ | $3.8 \times 10^{-4}$ |
| remove sparse | 0.37404 | 0.37398 | $5.0 \times 10^{-4}$ | $4.2 \times 10^{-4}$ |

From the result in Table 2, we can see that Calibrated Log Loss Metric has a smaller standard deviation (16% - 19% smaller) while the mean of Log Loss Metric and Calibrated Log Loss Metric is almost on par (within 0.02% difference).

**Pipelines with Different Model Architectures** In this set of experiments, our objective is to find out whether the new metric is capable of detecting a modeling improvement from changes in architecture. We tested a variety of different model architectures, including DCN (Wang et al., 2017), DeepFM (Guo et al., 2017), FNN (Zhang et al., 2016), and DCNMix (Wang et al., 2021).

Table 3: Accuracy of Log Loss Metric (LL) and Calibrated Log Loss Metric (CLL) (model architectures)

| Pipeline A | Pipeline B | LL Acc | CLL Acc |
|---|---|---|---|
| DCN | DCNMix | 64.4% | 71.5% |
| DeepFM | DCN | 77.2% | 83.9% |
| DeepFM | FNN | 76.9% | 79.9% |
| FNN | DCNMix | 61.5% | 72.0% |
| DeepFM | DCNMix | 84.8% | 93.4% |

Table 4: Mean and Standard Deviation of Log Loss Metric (LL) and Calibrated Log Loss Metric (CLL) (model architectures)

| Pipeline | LL Mean | CLL Mean | LL Std | CLL Std |
|---|---|---|---|---|
| DCN | 0.38021 | 0.38011 | $4.4 \times 10^{-4}$ | $3.3 \times 10^{-4}$ |
| DeepFM | 0.37971 | 0.3796 | $5.9 \times 10^{-4}$ | $3.7 \times 10^{-4}$ |
| FNN | 0.38029 | 0.38006 | $6.4 \times 10^{-4}$ | $4.0 \times 10^{-4}$ |
| DCNMix | 0.38046 | 0.38037 | $4.8 \times 10^{-4}$ | $3.4 \times 10^{-4}$ |

From the result in Table 3, we can clearly see that Calibrated Log Loss Metric has higher accuracy, again indicating its effectiveness when comparing the performance of pipelines with different model architectures. In Table 4, we report the mean and standard deviation of Log Loss Metric and Calibrated Log Loss Metric, consistent with previous results.

**Pipelines with Different Model Hyperparameters** In this set of experiments, we compare pipelines with different model hyperparameters, including neural network layer size, Batch Normalization (BN) (Ioffe & Szegedy, 2015), Dropout (Srivastava et al., 2014), and regularization weight.

In the first experiment, we compare a pipeline using the baseline model size with a pipeline using a smaller model size. In the second experiment, we compare a pipeline that uses batch normalization with a pipeline that does not use batch normalization. In the third experiment, we compare a pipeline that does not use Dropout with a pipeline that uses Dropout with dropout probability $0.7$. In the fourth experiment, we compare a pipeline that does not use regularization with a pipeline that uses L2 regularization with regularization weight $10^{-6}$.

Figure 1 illustrates the distribution of Log Loss Metric and Calibration Log Loss Metric observed in the Batch Normalization Experiments. We can clearly see that by using Calibrated Log Loss Metric, it becomes easier to separate pipeline with Batch Normalization from pipeline without Batch Normalization.

Table 5: Accuracy of Log Loss Metric (LL) and Calibrated Log Loss Metric (CLL) (hyperparameters)

| Pipeline A | Pipeline B | LL Acc | CLL Acc |
|---|---|---|---|
| Baseline Size | Smaller Size | 69.6% | 73.6% |
| BN | no BN | 80.2% | 89.7% |
| no Dropout | p = 0.7 | 95.0% | 99.3% |
| no regularization | weight $10^{-6}$ | 95.2% | 98.8% |

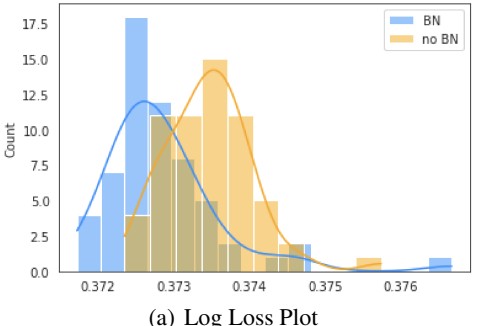

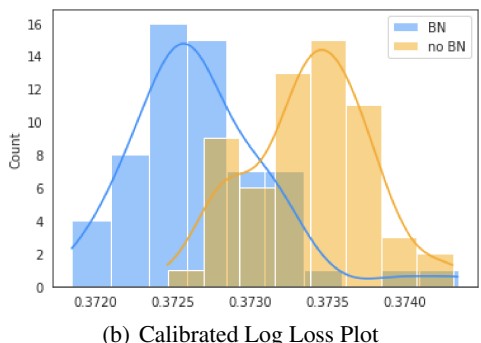

(a) Log Loss Plot        (b) Calibrated Log Loss Plot

Figure 1: Batch Normalization Experiment

From the result in Table 5, we can see that Calibrated Log Loss Metric has a higher accuracy regardless of the hyperparameters we are tuning, indicating its effectiveness when comparing the performance of pipelines with different hyperparameters, which is a very common task in Deep Learning. In Appendix B.3 Table 12, we report the mean and standard deviation of Log Loss Metric and Calibrated Log Loss Metric, again consistent with previous results.

**Pipelines with Different Levels of Regularization** In this set of experiments, we take a closer look at one hyperparameter we conduct in the previous section: regularization weight. For pipeline $A$, we use the baseline model. For pipeline $B$, we use different L2 regularization weights.

Table 6: Accuracy of Log Loss Metric (LL) and Calibrated Log Loss Metric (CLL) (regularization weight)

| Pipeline A | Pipeline B | LL Acc | CLL Acc |
|---|---|---|---|
| no regularization | weight $3 \times 10^{-7}$ | 63.2% | 69.3% |
| no regularization | weight $5 \times 10^{-7}$ | 82.2% | 88.2% |
| no regularization | weight $7 \times 10^{-7}$ | 86.6% | 92.4% |
| no regularization | weight $1 \times 10^{-6}$ | 95.2% | 98.8% |
| no regularization | weight $2 \times 10^{-6}$ | 98.8% | 100.0% |

From the result in Table 6, we can see that Calibrated Log Loss Metric has a higher accuracy across all different regularization weights, indicating its robustness to different values of regularization weight. As we increase the regularization weight in the pipeline $B$, the accuracies of both metrics increase. This is because pipelines $A$ and $B$ differ more with larger regularization weight, making performance comparison easier.

From the result in Appendix B.3 Table 13, we can see that Calibrated Log Loss Metric has a much smaller standard deviation (15% - 40% smaller) than Log Loss Metric while the mean of Log Loss Metric and Calibrated Log Loss Metric is almost on par (within 0.05% difference), again consistent with previous results.

## 5.4 CRITEO AD KAGGLE DATASET

The Criteo Ad Kaggle dataset[3] is another common benchmark dataset for CTR predictions. We provide additional experimental results in the Appendix B.4 using the deep learning recommendation model (DLRM) (Naumov et al., 2019) open-sourced by Meta as our baseline model. These results demonstrate that our method is generalized to different datasets and different models.

## 5.5 US STOCK MARKETS DATASET

**Dataset** For our experiments, we used per minute data from selected US stock equities over the period from January 2012 to December 2018, covering all available trading days. The raw input data consist solely of price information, including Open, Close, High, Low, and Volume (OCHLV). Our objective is to predict 10-minute returns using these data. Feature construction was carried out following the methodology provided in the Microsoft Qlib library (Yang et al., 2020)[4]. More details on the dataset are provided in the Appendix B.5.

**Base Model** We utilize Long Short-Term Memory (LSTM) networks (Hochreiter, 1997), as implemented in Yang et al. (2020), as our base model. To ensure that it has reasonable predictive power, we performed hyperparameter tuning. Data from 2012 to 2014 are used for training, while data from 2015 to 2018 serve as the validation set. During the testing phase within the validation data, to handle concept drift, we apply a 'Rolling Retraining' strategy, where the model is periodically fine-tuned by incorporating newly accumulated data every 6 months.

**Metrics** We compare the accuracy of Calibrated Quadratic Loss Metric and Quadratic Loss Metric. For each pipeline, we train the model 10 times with different initialization seeds and data shuffle to compute $\widehat{\text{Acc}}(\bar{e})$. We use the 'Quadratic Loss Metric' as the ground-truth metric for ranking the performance of different pipelines. To compute the Calibrated Quadratic Loss Metric, we applied a rolling strategy during the validation phase: the bias is calculated using the previous month's predictions and then is used to calibrate the following month's predictions.

**Pipelines with Different Levels of Regularization**

Table 7: Accuracy of Quadratic Loss Metric (QL) and Calibrated Quadratic Loss Metric (CQL) (regularization weight)

| Pipeline A | Pipeline B | QL Acc | CQL Acc |
|---|---|---|---|
| no regularization | weight $3 \times 10^{-6}$ | 58% | 76% |
| no regularization | weight $4 \times 10^{-6}$ | 76% | 86% |
| no regularization | weight $5 \times 10^{-6}$ | 77% | 91% |
| no regularization | weight $6 \times 10^{-6}$ | 87% | 94% |
| no regularization | weight $7 \times 10^{-6}$ | 92% | 96% |

From the result in Table 7, we can see that Calibrated Quadratic Loss Metric has a higher accuracy for all different regularization weights, similar to the result in Table 6. This indicates that our method generalizes to different loss metrics, different models, and different tasks.

**Additional Experimental Results** Appendix B.5 provides further experimental results, including tuning for hidden_size, num_layers, batch_size and num_epochs.

## 6 RELATED WORK

### 6.1 MODEL SELECTION IN SUPERVISED LEARNING SETTING

Model selection in supervised learning is crucial to ensure robust generalization to unseen data. A common approach is the *hold-out* method, which splits the data into three parts: training, validation,

---

[3]https://www.kaggle.com/c/criteo-display-ad-challenge
[4]https://github.com/microsoft/qlib/blob/main/qlib/contrib/data/handler.py

and test sets. The training set is used to learn the model, the validation set is employed for hyperparameter tuning, and the test set is reserved for final evaluation. Although straightforward, the hold-out method requires a large dataset to ensure that sufficient data are available for each subset.

In scenarios with limited data, *k-fold cross-validation* (Kohavi, 1995) is often preferred. This method splits the data into $k$ folds, trains the model in $k - 1$ folds, and validates it on the remaining fold, repeating this process $k$ times, and averaging the results. This ensures that each data point is used for both training and validation, providing a more comprehensive assessment of model performance. A special case of this method is *leave-one-out cross-validation*, where $k$ equals the number of training examples. However, k-fold cross-validation typically demands multiple model trainings, which can be impractical in deep learning due to the high computational cost.

More recently, Immer et al. (2021) proposed a scalable marginal-likelihood estimation method to select both hyperparameters and network architectures based on the training data alone. Their method is useful when validation data are unavailable. You et al. (2019) proposed Deep Embedded Validation method in Deep Unsupervised Domain Adaptation setting.

### 6.2 EVALUATING THE PERFORMANCE OF CTR PREDICTION MODELS

Evaluating the performance of CTR prediction models is crucial, with several metrics commonly used for this purpose (Yi et al., 2013). The Area Under the ROC Curve (AUC) (Fawcett, 2006; 2004) and its variants (Zhu et al., 2017), along with Log Loss Metric, are among the most prevalent metrics in this domain. For example, (He et al., 2014; Wang et al., 2017; McMahan et al., 2013) use Log Loss Metric as their core metric, while (Zhou et al., 2018; McMahan et al., 2013) use AUC as their core metric. However, AUC has been criticized for not taking into account the predictive probability (Yi et al., 2013). Log Loss Metric, in particular, is favored in scenarios that require calibrated predictions due to its consideration of predictive probabilities, an aspect crucial for applications such as Ads CTR predictions (He et al., 2014).

### 6.3 EVALUATING THE PERFORMANCE OF STOCK RETURN PREDICTION MODELS

In Hu et al. (2021), MSE (or equivalently RMSE) is identified as the most commonly used performance metric. Besides MSE, metrics such as MAPE, MAE, Accuracy, Sharpe Ratio, and Return Rate are also frequently used to evaluate model performance.

## 7 CONCLUSION AND DISCUSSION

**Conclusion** In this paper, we have presented a new approach to comparing the performance of different deep learning pipelines. We proposed a new metric framework, Calibrated Metric, which has higher accuracy and smaller variance than its vanilla counterpart for a wide range of tasks, models, and training pipelines. Our experiments in section 5 demonstrated the superiority of the Calibrated Metric, and we believe that this new metric can be used to compare the performance of different pipelines more effectively and efficiently. Future work includes expanding this idea to evaluate Natural Language Processing (NLP) and Computer Vision (CV) pipelines and establish theoretical guarantees in more general settings.

**Limitations** Our method sacrifices accuracy when comparing some specific pipelines. For example, if pipeline $B$ can reliably improve the model calibration over pipeline $A$, Calibrated Metric will not be able to correctly detect the benefits of pipeline B. However, for most pipeline comparisons conducted in industry and academia such as feature engineering, tuning parameters, etc., the Calibrated Metric has a boost in accuracy over its counterpart metric, as we demonstrated in Section 5.

**Potential Applications** Our method may have applications in the AutoML domain. AutoML (Automated Machine Learning) systems are designed to automate the process of selecting, designing, and tuning machine learning models, and a key component of these systems is the selection of the best-performing pipeline (e.g. hyperparameters, model architectures, etc.). The new metric could be used as a more accurate way to compare performance and select the best. The new metric is, in particular, useful when performing hyperparameter tuning.

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

## A   PROOFS

### A.1   PROOFS OF THEOREM 4.1 AND COROLLARY 4.2

**Lemma A.1.** *Suppose that $\hat{\beta}_n$ is the unique linear regression solution computed using the training data $\{X_i, Y_i\}_{i=1}^n$. Then, $\hat{\beta}_n$ is independent to $\{\bar{X}, \bar{Y}\}$, where*

$$\bar{X} = \frac{1}{n} \sum_{i=1}^n X_i, \text{ and } \bar{Y} = \frac{1}{n} \sum_{i=1}^n Y_i.$$

*Proof.* It is well-known that $\hat{\beta}_n$ is the solution of the convex program

$$\min_{\beta, c} \sum_{i=1}^n \left( Y_i - \beta^\top X_i - c \right)^2,$$

which is equivalent to the convex program

$$\min_\beta \sum_{i=1}^n \left( Y_i - \beta^\top X_i - \left( \bar{Y} - \beta^\top \bar{X} \right) \right)^2 \tag{9}$$

$$= \min_\beta \sum_{i=1}^n \left( \left( Y_i - \bar{Y} \right) - \beta^\top \left( X_i - \bar{X} \right) \right)^2.$$

Let $\tilde{Y}_i = Y_i - \bar{Y}$ and $\tilde{X}_i = Y_i - \bar{Y}$. Note that $\tilde{Y}_i$ is independent to $\bar{Y}$ and $\tilde{X}_i$ is independent to $\bar{X}$ as

$$cov(Y_i - \bar{Y}, \bar{Y}) = 0, cov(X_i - \bar{X}, \bar{X}) = 0,$$

and $\{X, Y\}$ are jointly normal. Note that the convex program (9) yields that $\hat{\beta}_n$ is a function of $\left\{ \tilde{X}_i, \tilde{Y}_i \right\}_{i=1}^n$, which is independent to $\{\bar{X}, \bar{Y}\}$. □

**Theorem A.2.** *Suppose that the features $X \in \mathbb{R}^d$ and the label $Y$ are distributed jointly Gaussian. We consider linear regression $h(x) = \beta^\top x + \alpha$. Let $\hat{\beta}_n$ be the coefficient learned from the training data with sample size $n$. Then, we have*

$$\left( 1 + \frac{1}{n} \right) \mathbb{E}[e_1(h(X), Y) | \hat{\beta}_n] = \mathbb{E}[e(h(X), Y) | \hat{\beta}_n],$$

*where the expectation is taken over the randomness over both the training and test samples.*

*Proof.* Note that the learned bias $\hat{\alpha} = \bar{Y} - \hat{\beta}_n^\intercal \bar{X}$, where $\bar{Y}$ and $\bar{X}$ are the empirical average of the samples in the training set. Then, the risks are defined as

$$\mathbb{E}\left[ e(h(X), Y) | \hat{\beta}_n \right] = \mathbb{E}\left[ \left( \left( Y - \bar{Y} \right) - \hat{\beta}_n^\intercal \left( X - \bar{X} \right) \right)^2 | \hat{\beta}_n \right],$$

$$\mathbb{E}\left[ e_1(h(X), Y) | \hat{\beta}_n \right] = \mathbb{E}\left[ \left( \left( Y - \mathbb{E}_\mathcal{D}[Y] \right) - \hat{\beta}_n^\intercal \left( X - \mathbb{E}_\mathcal{D}[X] \right) \right)^2 | \hat{\beta}_n \right]. \tag{10}$$

Therefore, we have

$$\mathbb{E}\left[ e_1(h(X), Y) | \hat{\beta}_n \right] = var(Y - \hat{\beta}_n^\intercal X | \hat{\beta}_n).$$

Note that we have

$$\{Y - \bar{Y}, X - \bar{X}\} \overset{d}{=} \sqrt{1 + \frac{1}{n}} \{Y - \mathbb{E}_\mathcal{D}[Y], X - \mathbb{E}_\mathcal{D}[X]\}.$$

given that $\{Y, X\}$ is independent to $\{\bar{Y}, \bar{X}\}$. Recall for A.1 that $\hat{\beta}_n$ is independent to $\{\bar{Y}, \bar{X}\}$, we have

$$\mathbb{E}\left[ e(h(X), Y) | \hat{\beta}_n \right] = var\left( \left( Y - \bar{Y} \right) - \hat{\beta}_n^\intercal \left( X - \bar{X} \right) | \hat{\beta}_n \right)$$

$$= \left( 1 + \frac{1}{n} \right) var(Y - \hat{\beta}_n^\intercal X | \hat{\beta}_n).$$

□

**Corollary A.3.** *Suppose that $h_1(x)$ and $h_2(x)$ are two different learned linear functions in different feature sets. Then, we have*

$$\mathbb{E}[R_e(h_1)] = \mathbb{E}[R_e(h_2)] \Leftrightarrow \mathbb{E}[R_{e_1}(h_1)] = \mathbb{E}[R_{e_1}(h_2)] \tag{11}$$

*and $var((1 + 1/n)\, R_{e_1}(h)) < var(R_e(h))$ for any $h$ learned from linear regression.*

*Proof.* From the definition, we see

$$\mathbb{E}[R_e(h)] = \mathbb{E}\left[\mathbb{E}\left[e(h(X), Y)|\hat{\beta}_n\right]\right],$$

$$\mathbb{E}[R_{e_1}(h)] = \mathbb{E}\left[\mathbb{E}\left[e_1(h(X), Y)|\hat{\beta}_n\right]\right].$$

Therefore, we conclude the first claim.

For the second claim, note that

$$R_{e_1}(h) = \mathbb{E}\left[e_1(h(X), Y)|\hat{\beta}_n\right],$$

$$R_e(h) = \mathbb{E}\left[e_1(h(X), Y)|\hat{\beta}_n, \bar{X}, \bar{Y}\right].$$

Then, the variance of $R_e(h)$ can be decomposed as

$$
\begin{aligned}
var(R_e(h)) &= var\left(\mathbb{E}\left[e(h(X), Y)|\hat{\beta}_n\right]\right) \\
&\quad + \mathbb{E}\left[var\left(\mathbb{E}\left[e(h(X), Y)|\hat{\beta}_n^{\mathsf{T}}, \bar{X}, \bar{Y}\right]|\hat{\beta}_n\right)\right] \\
&> var\left(\mathbb{E}\left[e(h(X), Y)|\hat{\beta}_n\right]\right) \\
&= var\left(\left(1 + \frac{1}{n}\right)\mathbb{E}\left[e_1(h(X), Y)|\hat{\beta}_n\right]\right) \\
&= var\left(\left(1 + \frac{1}{n}\right) R_{e_1}(h)\right).
\end{aligned}
$$

$\square$

# B EXPERIMENTS

## B.1 SYNTHETIC DATA: LINEAR REGRESSION

In this section, we consider a linear regression model to give empirical evidence to support our theory. We assume the response $Y$ follows the following generating process:

$$Y = \beta^\top X + \epsilon, \tag{12}$$

where $\epsilon \sim \mathcal{N}(\mu_e, \Sigma_e)$ and $\beta, X \in \mathbb{R}^d$.

In the experiments, we consider $d = 20$, $\beta = [1, 1, \ldots, 1]^\top$, and $X \sim \mathcal{N}(\mu_\mathcal{D}, \Sigma_\mathcal{D})$ in both the training set and the validation set. In the training set, we generate $N_{\text{train}} = 1000$ i.i.d. training samples to train a linear regression model. In the validation set, we generate $N_{\text{val}} = 11000$ i.i.d. samples, with $N_{\text{val}-\text{bias}} = 1000$ and $N_{\text{val}-\text{remain}} = 10000$.

We assume $\mu_\mathcal{D} = [-0.05, -0.05, \ldots, -0.05]^\top$, $\Sigma_\mathcal{D} = 0.25^2 \times I_{d \times d}$, $\mu_e = 1$ and $\Sigma_e = 2$.

Note that there is no randomness in the linear regression training process, as it is a convex optimization program. The randomness of Linear Regression comes from the training data. In order to run pipelines $A$ and $B$ multiple times to estimate the metric accuracy, we re-sample training data each time from the ground truth data distribution.

For pipeline $A$, we use all available 20 features, and for pipeline $B$, we use the first 19 features and leave the last one out. It is clear that the pipeline $A$ should perform better than the pipeline $B$ in the ground truth.

For each round of experiments, we run pipelines A and B for 100 times and report accuracy $\widehat{\mathrm{Acc}}(\bar{e})$ in Table 8. We performed 100 rounds of experiments and report the mean and standard errors of $\widehat{\mathrm{Acc}}(\bar{e})$ in Table 8. We also calculate the standard deviation and mean of Quadratic Loss (QL) Metric and Calibrated Quadratic Loss (CQL) Metric from pipeline A in each round of experiments, and report the average in Table 9.

Table 8: Accuracy of Quadratic Loss Metric (QL) and Calibrated Quadratic Loss (CQL) Metric under Linear Regression

| # of feature | | Accuracy | |
| --- | --- | --- | --- |
| Pipeline A | Pipeline B | QL | CQL |
| 20 | 19 | $93.5\% \pm 0.19\%$ | $94.53\% \pm 0.17\%$ |

Table 9: Mean and Standard Deviation of Quadratic Loss Metric (QL) and Calibrated Quadratic Loss (CQL) Metric under Linear Regression

| QL Mean | CQL Mean | QL Std | CQL Std |
| --- | --- | --- | --- |
| 4.067 | 4.069 | 0.0298 | 0.0286 |

From the result in Table 8, we can see that Calibrated Quadratic Loss Metric has a higher accuracy compared with Quadratic Loss Metric. From the result in Table 9, we can see that Calibrated Quadratic Loss Metric indeed has a smaller standard deviation than Quadratic Loss Metric while the mean of Quadratic Loss Metric and Calibrated Quadratic Loss Metric is almost on par.

### B.2 SYNTHETIC DATA: LOGISTIC REGRESSION

We consider a logistic regression model. We assume that the response $Y$ follows the Bernoulli distribution with probability $\left(1 + \exp(-\beta^\top X)\right)^{-1}$, for $\beta, X \in \mathbb{R}^d$.

In the experiments, we consider $d = 20$, $\beta = [1, 1, \ldots, 1]^\top$, and $X \sim \mathcal{N}(\mu_{\mathcal{D}}, \Sigma_{\mathcal{D}})$ in both the training and the validation sets. In the training set, we generate $N_{\mathrm{train}} = 1000$ i.i.d. training samples to train a logistic regression model. In the validation set, we generate $N_{\mathrm{val}} = 12000$ i.i.d. samples, with $N_{\mathrm{val-bias}} = 2000$ and $N_{\mathrm{val-remain}} = 10000$.

We assume $\mu_{\mathcal{D}} = [-0.05, -0.05, \ldots, -0.05]^\top$ and $\Sigma_{\mathcal{D}} = 0.25^2 \times I_{d \times d}$.

Note that similar to Linear Regression, there is no randomness in the training process of Logistic Regression as well, as it is a convex optimization program. The randomness of Logistic Regression comes from the training data. We employ the same strategy to estimate the metric accuracy, i.e. we re-sample training data each time from the ground truth data distribution.

For pipeline $A$, we use all available 20 features, and for pipeline $B$, we use the first 19 features and leave the last one out. It is clear that the pipeline $A$ should perform better than the pipeline $B$ in the ground truth.

For each round of experiments, we run pipelines A and B for 100 times and report accuracy $\widehat{\mathrm{Acc}}(\bar{e})$ in Table 10. We performed 100 rounds of experiments and report the mean and standard errors of $\widehat{\mathrm{Acc}}(\bar{e})$ in Table 10. We also calculate the standard deviation and mean of Log Loss (LL) Metric and Calibrated Log Loss (CLL) Metric from pipeline A in each round of experiments and report the average in Table 11.

From the result in Table 10, we can clearly see that Calibrated Log Loss Metric has a huge accuracy boost compared with Log Loss Metric. From the result in Table 11, we can see that Calibrated Log Loss Metric indeed has a smaller standard deviation than Log Loss Metric while the mean of Log Loss Metric and Calibrated Log Loss Metric is almost on par.

Table 10: Accuracy of Log Loss Metric (LL) and Calibrated Log Loss Metric (CLL) under Logistic Regression

| # of feature | | Accuracy | |
|---|---|---|---|
| Pipeline A | Pipeline B | LL | CLL |
| 20 | 19 | $85.93\% \pm 0.26\%$ | $89.36\% \pm 0.24\%$ |

Table 11: Mean and Standard Deviation of Log Loss Metric (LL) and Calibrated Log Loss Metric (CLL) under Logistic Regression

| LL Mean | CLL Mean | LL Std | CLL Std |
|---|---|---|---|
| 0.5249 | 0.5219 | 0.00376 | 0.00357 |

### B.3 AVAZU CTR PREDICTION DATASET

We report the mean and standard deviation of Log Loss Metric and Calibrated Log Loss Metric for additional experiments in the Avazu CTR Prediction dataset.

Table 12: Mean and Standard Deviation of Log Loss Metric (LL) and Calibrated Log Loss Metric (CLL) (hyperparameters)

| Pipeline | LL Mean | CLL Mean | LL Std | CLL Std |
|---|---|---|---|---|
| Baseline | 0.37347 | 0.37338 | $5.8 \times 10^{-4}$ | $3.8 \times 10^{-4}$ |
| smaller size | 0.37383 | 0.37374 | $4.8 \times 10^{-4}$ | $3.9 \times 10^{-4}$ |
| BN | 0.37286 | 0.37268 | $7.8 \times 10^{-4}$ | $4.4 \times 10^{-4}$ |
| Dropout | 0.37454 | 0.37456 | $3.9 \times 10^{-4}$ | $3.6 \times 10^{-4}$ |
| Regularization | 0.37475 | 0.37459 | $6.1 \times 10^{-4}$ | $4.2 \times 10^{-4}$ |

Table 13: Mean and Standard Deviation of Log Loss Metric (LL) and Calibrated Log Loss Metric (CLL) (regularization weight)

| Pipeline | LL Mean | CLL Mean | LL Std | CLL Std |
|---|---|---|---|---|
| 0 | 0.37347 | 0.37338 | $5.8 \times 10^{-4}$ | $3.8 \times 10^{-4}$ |
| $3 \times 10^{-7}$ | 0.37371 | 0.37369 | $4.8 \times 10^{-4}$ | $4.3 \times 10^{-4}$ |
| $5 \times 10^{-7}$ | 0.37419 | 0.37411 | $5.9 \times 10^{-4}$ | $5.0 \times 10^{-4}$ |
| $7 \times 10^{-7}$ | 0.37428 | 0.37421 | $5.7 \times 10^{-4}$ | $4.4 \times 10^{-4}$ |
| $1 \times 10^{-6}$ | 0.37475 | 0.37459 | $6.1 \times 10^{-4}$ | $4.2 \times 10^{-4}$ |
| $2 \times 10^{-6}$ | 0.37562 | 0.37547 | $5.9 \times 10^{-4}$ | $4.2 \times 10^{-4}$ |

### B.4 CRITEO AD KAGGLE DATASET

**Dataset** The Criteo Ad Kaggle dataset is a common benchmark dataset for CTR predictions. It consists of a week's worth of data, approximately 45 million samples in total. Each data point contains a binary label, which indicates whether the user clicks or not, along with 13 continuous, 26 categorical features. The positive label accounts for $25.3\%$ of all data. The categorical features consist of 1.3 million categories on average, with 1 feature having more than 10 million categories, 5 features having more than 1 million categories. Due to computational constraints in our experiments, we use the first 15 million samples, shuffle the dataset randomly, and split the whole dataset into $85\%$ $\hat{\mathcal{D}}_{\text{train}}$, $3\%$ $\hat{\mathcal{D}}_{\text{val-bias}}$, $12\%$ $\hat{\mathcal{D}}_{\text{val-remain}}$.

**Metrics** We compare the accuracy of Calibrated Log Loss Metric and Log Loss Metric. To make a fair comparison, we compute Log Loss metric on $\hat{\mathcal{D}}_{\text{val}} = \hat{\mathcal{D}}_{\text{val-bias}} + \hat{\mathcal{D}}_{\text{val-remain}}$.

**Base Model 1** We use the deep learning recommendation model (DLRM) (Naumov et al., 2019) open-sourced by Meta as our baseline model. DLRM employs a standard architecture for ads recommendation tasks, with embeddings to handle categorical features, Multilayer perceptrons (MLPs) to handle continuous features and the interactions of categorical features and continuous features. Throughout our experiments, we use the default parameters and a SGD optimizer. For each pipeline, we train the model 25 times with different initialization seeds and data orders to calculate $\widehat{\text{Acc}}(\bar{e})$.

**Base Model 2** We use the open source xDeepFM model (Lian et al., 2018) implemented in Shen (2017) as our base model, same as Avazu CTR Prediction dataset. For each pipeline, we train the model 60 times with different initialization seeds and data orders to calculate $\widehat{\text{Acc}}(\bar{e})$.

**Pipelines with Different Embedding Dimension (Base Model = DLRM)**

Table 14: Accuracy of Log Loss Metric (LL) and Calibrated Log Loss Metric (CLL) (Embedding Dimension)

| Pipeline A | Pipeline B | LL Acc | CLL Acc |
|---|---|---|---|
| Embedding Dimension = 14 | Embedding Dimension = 16 | 55.4% | 60.3% |
| Embedding Dimension = 14 | Embedding Dimension = 18 | 61.9% | 70.4% |
| Embedding Dimension = 16 | Embedding Dimension = 18 | 61.8% | 65.8% |

**Pipelines with Different Batch Size (Base Model = DLRM)**

Table 15: Accuracy of Log Loss Metric (LL) and Calibrated Log Loss Metric (CLL) (Batch Size)

| Pipeline A | Pipeline B | LL Acc | CLL Acc |
|---|---|---|---|
| Batch Size = 256 | Batch Size = 128 | 84.5% | 88.0% |
| Batch Size = 512 | Batch Size = 128 | 89.0% | 95.5% |

**Pipelines with Different MLP Width (Base Model = DLRM)**

Table 16: Accuracy of Log Loss Metric (LL) and Calibrated Log Loss Metric (CLL) (MLP Width)

| Pipeline A | Pipeline B | LL Acc | CLL Acc |
|---|---|---|---|
| MLP = 128-256-1 | MLP = 256-256-1 | 57.4% | 60.2% |
| MLP = 128-256-1 | MLP = 512-256-1 | 60.5% | 63.4% |
| MLP = 128-256-1 | MLP = 1024-256-1 | 58.6% | 61.4% |

**Pipelines with Different Number of Features (Base Model = xDeepFM)**

Table 17: Accuracy of Log Loss Metric (LL) and Calibrated Log Loss Metric (CLL) (features)

| Pipeline A | Pipeline B | LL Acc | CLL Acc |
|---|---|---|---|
| Baseline | remove dense | 96.7% | 100.0% |
| Baseline | remove sparse | 87.6% | 95.7% |

**Pipelines with Different Model Hyperparameters (Model = xDeepFM)**

**Pipelines with Different Levels of Regularization (Model = xDeepFM)**

B.5  US STOCK MARKETS DATASET

**Dataset** For our experiments, we used per minute data from selected US stock equities over the period from January 2012 to December 2018, covering all available trading days. The raw input

Table 18: Accuracy of Log Loss Metric (LL) and Calibrated Log Loss Metric (CLL) (hyperparameters)

| Pipeline A | Pipeline B | LL Acc | CLL Acc |
|---|---|---|---|
| Baseline Size | Smaller Size | 80.1% | 86.4% |
| no Dropout | p = 0.1 | 78.3% | 87.2% |

Table 19: Accuracy of Log Loss Metric (LL) and Calibrated Log Loss Metric (CLL) (regularization weight)

| Pipeline A | Pipeline B | LL Acc | CLL Acc |
|---|---|---|---|
| no regularization | weight $7 \times 10^{-2}$ | 78.1% | 84.4% |
| no regularization | weight $1 \times 10^{-1}$ | 82.3% | 90.0% |
| no regularization | weight $1.5 \times 10^{-1}$ | 90.0% | 96.4% |
| no regularization | weight $2 \times 10^{-1}$ | 91.0% | 98.1% |

data consist solely of price information, including Open, Close, High, Low, and Volume (OCHLV). Each year, there are approximately 250 active trading days with regular trading hours. The data were back-adjusted for all dividend payments received and stock splitting that occurred during this time. On an active day, the regular trading hours start at 09:30 am and close at 16:00 pm. However, stock exchanges are closed all day during weekends and official holidays, and no trading is permitted. Some official holidays are New Year's Day, Good Friday, Independence Day, and Christmas Day. The stock exchanges close early on a few days of the year, operating only from 09:30 am to 13:00 pm, e.g., the day before Independence Day, Christmas Eve, for example. The complete list of holidays and early closings is obtained from algoseek[56] to pre-process the data and generate precise time series. The barchart[7] is the source of price data.

**Selected US Stock Equities** We selected the following 25 stocks for our experiments based on the data quality, specifically due to fewer missing rows. The selected stocks are: DIS, VZ, JPM, GE, COP, F, MSFT, XOM, AAPL, CMCSA, WFC, PFE, INTC, GILD, CVX, BAC, GM, SBUX, T, C, QCOM, JNJ, WMT, KO, CSCO.

**Pipelines with Different hidden_size**

Table 20: Accuracy of Quadratic Loss Metric (QL) and Calibrated Quadratic Loss Metric (CQL) (hidden_size)

| Pipeline A | Pipeline B | QL Acc | CQL Acc |
|---|---|---|---|
| hidden_size = 32 | hidden_size = 64 | 69% | 76% |
| hidden_size = 64 | hidden_size = 96 | 74% | 78% |
| hidden_size = 64 | hidden_size = 112 | 88% | 90% |
| hidden_size = 64 | hidden_size = 128 | 91% | 94% |
| hidden_size = 64 | hidden_size = 144 | 97% | 99% |

**Pipelines with Different num_layers**

**Pipelines with Different batch_size**

**Pipelines with Different num_epochs**

---

[5] https://us-equity-market-holidays.s3.amazonaws.com/holidays.csv
[6] https://us-equity-market-holidays.s3.amazonaws.com/earlycloses.csv
[7] https://www.barchart.com/

Table 21: Accuracy of Quadratic Loss Metric (QL) and Calibrated Quadratic Loss Metric (CQL) (num_layers)

| Pipeline A | Pipeline B | QL Acc | CQL Acc |
|---|---|---|---|
| num_layers = 3 | num_layers = 2 | 93% | 99% |
| num_layers = 4 | num_layers = 2 | 94% | 100% |
| num_layers = 4 | num_layers = 3 | 74% | 88% |
| num_layers = 5 | num_layers = 3 | 80% | 98% |
| num_layers = 6 | num_layers = 3 | 78% | 100% |
| num_layers = 6 | num_layers = 4 | 64% | 76% |

Table 22: Accuracy of Quadratic Loss Metric (QL) and Calibrated Quadratic Loss Metric (CQL) (batch_size)

| Pipeline A | Pipeline B | QL Acc | CQL Acc |
|---|---|---|---|
| batch_size = 3000 | batch_size = 2000 | 75% | 85% |
| batch_size = 3000 | batch_size = 1000 | 89% | 97% |
| batch_size = 2000 | batch_size = 1000 | 60% | 77% |

Table 23: Accuracy of Quadratic Loss Metric (QL) and Calibrated Quadratic Loss Metric (CQL) (num_epochs)

| Pipeline A | Pipeline B | QL Acc | CQL Acc |
|---|---|---|---|
| num_epochs = 4 | num_epochs = 5 | 70% | 75% |
| num_epochs = 4 | num_epochs = 6 | 94% | 96% |
| num_epochs = 4 | num_epochs = 7 | 97% | 99% |
| num_epochs = 5 | num_epochs = 6 | 98% | 98% |
| num_epochs = 5 | num_epochs = 7 | 99% | 100% |
| num_epochs = 6 | num_epochs = 7 | 72% | 82% |

