# OpenReview forum: "Towards Accurate Deep Learning Model Selection: A Calibrated Metric Approach"
_ICLR.cc/2025/Conference — ICLR 2025 Conference Withdrawn Submission_

### Official Review · Reviewer_1fBb · 2024-11-01

**Soundness:** 2
**Presentation:** 2
**Contribution:** 2
**Rating:** 3
**Confidence:** 3

**Summary:**

This paper address the question of how to more accurately compare two model pipelines given the randomness involved in producing models (e.g. random seed, stochastic optimization, etc.). The proposed method is to set aside some part of the validation data for "bias correction", and then compare two models with this correction using the original metric of interest on the remaining validation data. The authors demonstrate the efficacy of the proposed approach over several experimental settings.

**Strengths:**

- **Significance:** This paper addresses an increasingly important problem: how to compare two modeling approaches without repeated training runs over different random initializations. The authors put forth a methodology for doing so and conduct empirical verifications of their approach across various models, datasets, and hyperparameter settings.

- **Quality:** Overall the ideas in the paper are sound; the motivation makes sense, the experiments are described well, and there is some theoretical justification in a toy setting.

- **Clarity:** The flow of the paper is good -- the key problem is introduced thoroughly and the proposed framework as well as how to implement it in practice are described well.

- **Originality:** The idea of correcting model biases using some small amount of held-out data is not new and has appeared in many other parts of the literature (for example, post-training calibration); that being said, to my knowledge the use of this technique to reduce the variance in the comparison between two model pipelines does feel new and useful.

**Weaknesses:**

The main weakness I see with this work is the details of implementing this framework in arbitrary settings, and whether the justifications in the paper for the framework actually translate to such settings. My worry is that this paper essentially produces a modification to standard loss functions for evaluating testing performance, but we do not know in practice if this modification is actually getting better at measuring the true attribute we care about (hierarchy of models). Additionally, I am also concerned by the lack of comparisons to more recent related work.

**Framework Details:** Algorithm 1, in my view, is too high-level to be immediately used for the very general setting this paper tries to cover. In particular, the key aspect of this algorithm is the bias correction function $f_e$, but there is no procedure or even intuition for how one should think about this function. The examples regarding log loss and MSE are very helpful and make sense -- we just introduce a single parameter that adjusts the logits/outputs, but at the very least this notion deserves more explicit exposition at the start of Section 3. Another important aspect is how to actually choose the split of the validation data for fitting the bias correction vs. evaluation; the paper uses 2% for fitting and the remaining 18% for evaluation but this seems arbitrary. A discussion of the trade-off here would be useful. Lastly, as more of a pedantic point, in the experiments it appears that *val* data corresponds to *test* data, so it would also be useful to clarify terminology.

**Applicability in Practice:** The applicability of the proposed framework rests on the idea that the calibrated metric $\bar{e}$ has the same behavior as the original metric $e$ (in the sense that they order different models the same way when taking into account the ground truth distribution) but lower estimation variance. However, there is no justification that this is actually the case in the practical verifications; the theory in the paper applies only to linear regression with jointly Gaussian data, which is very much a toy setting. The experiments compare the "accuracy" (in the sense defined in the paper) of the calibrated metric $\bar{e}$ to that of the original metric $e$, but this comparison only makes sense if we know a priori that one modeling pipeline is better than another (in which case we are checking whether $\bar{e}$ uncovers the better model more accurately than $e$). I understand that intuitively we do have a sense of which pipeline should be better (since we a priori know that some features/training techniques make a nontrivial difference in the settings being considered), but given the generality of the proposed framework I don't see how the comparisons between $\bar{e}$ and $e$ can be conducted in arbitrary settings. For example, we might get better "accuracy" with \bar{e} in some setting but it is actually picking out the worse model (from the perspective of the test distribution).

**Related Work:** This paper has a very limited comparison to recent related work, which is concerning given that there has been a decent amount of work in this direction over the past few years. At the very least, this paper should compare to the ideas in [1] and the related work cited therein, as [1] also analyzes the variance of metrics computed on held-out test data and what conclusions can be drawn from comparing test metrics across different random training runs.

In summary, while I find the ideas in the paper to be interesting and the problem being considered to be important, I think stronger justifications (and details) are necessary to justify acceptance of the proposed method so I lean reject (falling somewhere between a 3 and a 5). I am happy to update my score upon engaging with the authors if they can clear up some of the issues pointed out above.

[1] Jordan, Keller. “On the Variance of Neural Network Training with respect to Test Sets and Distributions.” International Conference on Learning Representations (2023).

**Questions:**

- My main question is pointed out in the second weakness above: how do we know that higher accuracy in the sense of the paper is actually good in practice? A simple (counter)example would be some metric $\bar{e}$ that is always 0 for some model A and 1 for all other models; when comparing against model A this metric will have perfect accuracy since it always prefers model A, but it is not necessarily the case that model A is better than model B in some general setting.

---

### Official Review · Reviewer_gLLZ · 2024-11-02

**Soundness:** 2
**Presentation:** 3
**Contribution:** 3
**Rating:** 5
**Confidence:** 3

**Summary:**

This paper tackles the challenge of accurately comparing machine learning models. The authors propose adding a bias correction term to model predictions. This adjustment centers predictions around expected values, aiming to reduce reliance on sample-specific means. The authors claim that this approach leads to more accurate comparisons across different models and validation sets. They support this method with a rigorous theoretical analysis focused on linear models, showing how bias correction can reduce variance in model evaluations. Experiments demonstrate the approach’s effectiveness by comparing different model setups under two main configurations. Results show that the proposed calibrated log loss offers a more stable and reliable measure of model performance than standard log loss.

**Strengths:**

1. This paper addresses a critical problem in model evaluation, making comparisons more reliable for practical model selection.

2. The paper offers rigorous theoretical analysis for linear models.

3. Experiments show that the calibrated log loss outperforms standard metrics significantly and consistently.

**Weaknesses:**

1. To my understanding, biased mean predictions happen when examples are infinite. However, since the validation set (specifically $D_{val-bias}$ is typically smaller than the training set, why can we expect bias correction with $D_{val-bias}$ is more reliable than the training set when applying the model to unseen test data?

2. It appears that the experiments assume Pipeline A should outperform Pipeline B, but I found no clear explanation supporting this assumption in Tables 3, 5, 6, and 7. Without justification, it's difficult to interpret the effectiveness of the metrics in distinguishing between the pipelines.

3. Does the proposed method support multiclass classification problems? For binary classification, is this method particularly suitable for CTR or generally applicable to other datasets?

4. The term “calibrated” here is a bit ambiguous as it usually refers to confidence calibration in the context of deep learning. Clarifying the exact meaning will be helpful.

**Questions:**

See Weaknesses 1-3.

---

### Official Review · Reviewer_nNgX · 2024-11-03

**Soundness:** 1
**Presentation:** 1
**Contribution:** 1
**Rating:** 3
**Confidence:** 3

**Summary:**

This paper proposes the calibrated loss to compare different evaluation pipelines. The proposed calibrated loss is easy to compute and has favorable properties for comparing different pipelines, e.g., the variance of the metric is smaller than that of the non-calibrated one. Through numerical experiments, the performance of the proposed calibrated loss was demonstrated.

**Strengths:**

- The proposed calibrated loss is computed efficiently.

**Weaknesses:**

- The proposed calibrated loss is theoretically supported only under the assumption of linear regression. It is unclear whether the proposed loss is theoretically valid with other models, such as deep neural networks or decision trees.
- The experiments compared the non-calibrated and calibrated losses, but the final performance, i.e., standard accuracy evaluated by the 0-1 loss, is not reported. Thus, it is unclear if the model selected by the proposed calibrated loss will be the best.
- The comparison with the other model selection methods and/or the discussion about related work are missing. For example, the method can be compared with cross-validation and information criteria from the viewpoint of the model selection.

**Questions:**

- What is the standard accuracy or 0-1 error in the experiments? Usually, the surrogate loss, such as the log loss, is used in training, and the 0-1 loss is used in testing. The current results show the comparison based on the surrogate loss with and without calibration. However, since the final performance is evaluated by the standard accuracy (1 - the 0-1 error) in real-world applications, the connections between the proposed calibrated loss and such a standard accuracy are essential to demonstrate the effectiveness of the proposed calibrated loss.
- The results of the experiments show that the best hyperparameters of the non-calibrated and calibrated losses are the same. Are there any examples showing that the best models selected by the two losses are different?



Minor comments

- $f_e$ in L211 is not used. The equation in L197 with the different $e$ probably used $f_e$ for CQL in the experiments. If so, the explanation needs to be added.

---

### Official Review · Reviewer_2vr6 · 2024-11-04

**Soundness:** 3
**Presentation:** 2
**Contribution:** 2
**Rating:** 3
**Confidence:** 4

**Summary:**

The paper proposes to devote a part of the valid set for tuning a bias term added to the predictor, so that the predictor is better calibrated, before using it to evaluate the original metric on the remainder of the set. Doing this is shown to yield a more reliable model selection criterion compared to computing the metric using the initial ill-calibrated model, on the task of Click-Through-Rate prediction with the Crieto-ad dataset, and on a stock market return prediction task.

**Strengths:**

- Originality and significance: To my knowledge this paper is the first to highlight an interesting and potentially important connection between basic model miscalibration and an increase in variance and unreliability of evaluation metrics for doing model selection.

- Quality: The empirical evaluation is thorough and convincingly shows the benefit of using metrics post-calibration, for different kinds of model selections, on the two considered real data tasks.

- Clarity: The paper is overall well written and easy to follow. The principle is well-motivated and clearly explained.

**Weaknesses:**

- The work is not properly put in context of the research on deep learning model calibration. The Related Works section is extremely light, overly generic to the point of being barely relevant, and oblivious of research on model calibration, when the paper is essentially leveraging a basic calibration technique. See e.g. [1,2,3] as starting pointers.
Relatedly, the recalibration technique proposed for binary classification (Eq 6) is given ad-hoc without a reference.

- The theoretical analysis is very limited (linear regression only with joint Gaussian assumption) and somewhat superficial. It assumes perfect knowledge of target E[Y] and does not account for the fact that we have a finite validation set. Finite validation induces a noisy recalibration, moreover having sacrificed part of the set for calibration will yield additional variance in the estimation of the expected risk. None of these effects are accounted for.

- I find the work oversells basic recalibration as a new metric, instead of focusing on the core problems with miscalibration: why miscalibrated models shouldn’t be used for model selection (the core of the work’s contribution), and why we learn such miscalibrated models in the first place.

- Limited scope of the exposition (to binary classification and regression) and empirical evaluation. The approach is only relevant to the extent that the training pipeline produces miscalibrated models (in the limited sense of failing to match target averages). It would be more convincing if the authors had shown that the problem and remedy are relevant more broadly, beyond the two specific tasks studied, s.a. including established deep learning multiclass image classification benchmarks and pipelines.

**Questions:**

Q1: The recalibration technique proposed for binary classification (Eq 6) is given ad-hoc without a reference. It seems to correspond to a simple version of Platt scaling or temperature scaling, no? (See [1, 4]).

Q2: The success of your approach is directly related to how poorly calibrated the initial trained model ended up being. It would not bring any benefit if trained models are already well calibrated.
This begs the question: is there a benefit to selecting an uncalibrated model (based on the performance of its recalibrated version) rather than selecting the recalibrated version itself? If not, why not argue for training calibrated models, rather than for using the calibrated metric to select among uncalibrated models.

Q3: It would be interesting to define and include in your tables a measure of the “miscalibration” of learned models (what could it be?), in the limited sense you use of mismatch of the expectations, and assess whether that measure is correlated with the observed drop in model selection accuracy.

Q4: Why do you think we learn miscalibrated models (in the sense of mismatched expectations) in the first place?
Your intuition seems rooted in the volatility of the trained bias term. Is there anything else you think is responsible?
Can you think of ways to measure that volatility?
Ways to control and reduce that volatility?

Minor suggestions for improving the clarity of the paper:
- Define/explain your notations before use: $f_e$, its domain and range. Set or range of values for y. The reader shouldn’t have to guess.
- Algorithm 1 could be more detailed (e.g. clarifying how partition is done in step 3, and referring to the text for how to compute c* in step 5).
- Table captions could be more detailed.
- References section should have complete references and not use truncated author lists with “et al.” (to fix for papers “Accounting for variance in …”, “Practical lessons …”, and “Ad click prediction …”).

[1] Guo, C., Pleiss, G., Sun, Y., and Weinberger, K. Q. On calibration of modern neural networks. In International Conference on Machine Learning, 2017.

[2] Deng-Bao Wang, Lei Feng, and Min-Ling Zhang. Rethinking calibration of deep neural networks: Do not be afraid of overconfidence. NeurIPS, 2021.

[3] Calibration in Deep Learning: A Survey of the State-of-the-Art,
Cheng Wang 2023  https://arxiv.org/abs/2308.01222

[4] John Platt. Probabilistic outputs for support vector machines and comparisons to regularized likelihood methods. Advances in large margin classifiers, 1999.

---

### Official Review · Reviewer_EiBw · 2024-11-09

**Soundness:** 2
**Presentation:** 3
**Contribution:** 2
**Rating:** 3
**Confidence:** 3

**Summary:**

This paper formulates the deep learning pipeline evaluation and proposes a new metric called "Calibrated Metric" to overcome this problem. The authors devise 2 different metrics for binary classification and regression. A theoretical analysis on linear regression and experiments is provided to validate the effectiveness of the proposed metrics.

**Strengths:**

This paper has several strengths:
- This paper considers the problem of comparing deep learning pipelines which is overlooked in the literature.
- Empirical results show the superiority of the proposed metrics over the log-loss and mse.

**Weaknesses:**

This paper has several weaknesses:
- I wonder about the motivation of this problem. As far as I understand, the model $h$ in Alg.1 is trained before the alg.1. If it is the case, why did we need a new metric for pipeline comparison since the naive baseline is compare the accuracy on the validation/testing dataset.
- The theoretical result is only for linear regression, which seems trivial in the current practical setting where over-parameterized models are prevalent.
- This corrected bias is based on the choice of validation set. In the case of datasets where the validation set is not provided in advance, randomly selected $ \mathcal D_{val-bias}$ may cause variance in results. Therefore, I wonder about the robustness of this metric with different selected validation sets.
- Why were all pairs of models included in Table 3?
- For related work, there is another line of work called "transferability estimation" which aims to compare different checkpoints/tasks for a specific dataset. Therefore, the literature review would be more comprehensive if authors compared with this line of work.

**Questions:**

See the weaknesses

---

### Note · Authors · 2024-11-25

I have read and agree with the venue's withdrawal policy on behalf of myself and my co-authors.